# Effect of SARS-CoV-2 Infection and BNT162b2 Vaccination on the mRNA Expression of Genes Associated with Angiogenesis

**DOI:** 10.3390/ijms242216094

**Published:** 2023-11-08

**Authors:** Paulina Wigner-Jeziorska, Edyta Janik-Karpińska, Marta Niwald, Joanna Saluk, Elżbieta Miller

**Affiliations:** 1Department of General Biochemistry, Faculty of Biology and Environmental Protection, University of Lodz, 90-136 Lodz, Poland; paulina.wigner@biol.uni.lodz.pl (P.W.-J.); joanna.saluk@biol.uni.lodz.pl (J.S.); 2Biohazard Prevention Centre, Faculty of Biology and Environmental Protection, University of Lodz, 90-136 Lodz, Poland; edyta.janik@biol.uni.lodz.pl; 3Department of Neurological Rehabilitation, Medical University of Lodz, 90-136 Lodz, Poland; marta.niwald@umed.lodz.pl

**Keywords:** COVID-19, angiogenesis, mRNA expression, BNT162b2 vaccination

## Abstract

Severe acute respiratory syndrome coronavirus 2 (SARS-CoV-2), discovered in December 2019 in Wuhan, China, caused the coronavirus disease 2019 (COVID-19). Due to the rate of spread of this virus, the World Health Organization, in March 2020, recognised COVID-19 as a worldwide pandemic. The disease is multisystemic with varying degrees of severity. Unfortunately, despite intensive research, the molecular changes caused by SARS-CoV-2 remain unclear. Mechanisms affected by the virus infection include endothelial dysfunction and angiogenesis. Similarly, the vaccines developed so far affect the process of angiogenesis, contributing to the development of undesirable effects on part of the cardiovascular system. The presented research aimed to investigate the impact of the SARS-CoV-2 infection and the Pfizer Comirnaty vaccine (BNT162b2) on the molecular aspect of angiogenesis. We found that convalescents vaccinated with one dose of BNT162b2 were characterised by higher *MMP-7* (metalloproteinases 7) expression than non-vaccinated convalescents and healthy volunteers vaccinated with one dose of BNT162b2. Moreover, non-vaccinated convalescents showed increased mRNA expression of *ADAMTS1* (ADAM metallopeptidase with thrombospondin type 1 motif 1) compared to healthy volunteers vaccinated with one dose of BNT162b2. In addition, we showed significant sex differences in the expression of *MMP-7*. In conclusion, the results of our study suggest a significant impact of SARS-CoV-2 infection and vaccination on the course of angiogenesis at the molecular level.

## 1. Introduction

Although more than three years have passed since the World Health Organization (WHO) recognised the coronavirus disease 2019 (COVID-19) epidemic as a pandemic, the disease is still considered one of the most critical issues in healthcare systems around the world. Between the first unexplained pneumonia cases reported in Wuhan (China) and the time this manuscript was written, over 771 million cases of SARS-CoV-2 (severe acute respiratory syndrome coronavirus 2) infection had been recorded in 192 countries and territories. Out of this number, there are nearly 21 million active cases, over 66 million recoveries, and over 6.96 million deaths. This disease, with symptoms ranging from asymptomatic to critical in patients, was observed in different age categories [1]. Pneumonia, which can progress to acute respiratory distress syndrome (ARDS), is the most commonly observed in this viral infection [2]. However, in addition to pulmonary symptoms, patients also reported a myriad of extrapulmonary symptoms, including dizziness, skin symptoms, arrhythmia, liver damage, haematuria, and even loss of taste and smell. Therefore, COVID-19 is considered a highly heterogeneous and multi-system disease [3,4,5,6]. Some complications of COVID-19 disease are not a direct consequence of a viral infection but may be the result of global immune dysregulation caused by the virus [7]. In clinical studies, it has been shown that the most severe cases are characterised by elevated levels of pro-inflammatory cytokines, triggering a cytokine storm that is directly related to the progression of COVID-19 [8,9] and other biological responses such as extracellular neutrophil traps (NETs) or the production of the inflammatory diseases which, if they persist, can cause tissue damage [10,11]. The cytokine storms observed in COVID-19 are induced by the interaction between infected cells and cells of the host’s immune system, affecting many organs, even those without active viral replication [8,12,13,14]. Interestingly, the cytokine storm associated with the development of COVID-19 may also contribute to disorders of epithelial haemostasis accompanied by loss of balance of the renin–angiotensin–aldosterone system (RAAS), antithrombotic and immune functions. High levels of cytokines resulting from the cytokine storm contribute to changes in endothelial cells towards a chemotactic phenotype with high permeability and prothrombotic features. Moreover, the innate immune response is also responsible for oxidative damage to the endothelium due to TLRs (toll-like receptors) activation via DAMPs (danger/damage-associated molecular patterns) and PAMPs (pathogen-associated molecular patterns). These changes, along with RAAS abnormalities, are associated with a hypercoagulable state. SARS-CoV-2 infection leads to the loss of ACE2 (angiotensin-converting enzyme 2) activity in endothelial cells, which reduces the metabolism and inactivation of angiotensin II and consequently also reduces the level of angiotensin 1–7 (a by-product of angiotensin II degradation). Higher levels of angiotensin II combined with lower levels of angiotensin 1–7 cause vasoconstriction and adhesion of leukocytes and platelets, thus promoting thrombogenic effects and inhibiting fibrinolytic activity. These changes at the molecular level are reflected in disturbances in clinical parameters. Patients with COVID-19 are characterised by increased levels of fibrinogen, D-dimer and fibrin degradation products, von Willebrand factor/factor VIII, and lower levels of plasminogen activator inhibitor 1 [15,16,17,18,19]. Consequently, this hypercoagulable state associated with SARS-CoV-2 infection may lead to the development of cardiovascular complications in COVID-19 patients, such as myocardial damage and dysfunction, deep vein thrombosis and pulmonary thromboembolism. An important feature of COVID-19-related coagulopathy is damage to the microvascular endothelium in the pulmonary vascular beds, mainly due to the cytokine storm. It is believed that, similarly to acute respiratory distress syndrome, in the course of COVID-19, inflammatory markers (e.g., interleukin 1; IL-1) and neutrophil mediators (e.g., reactive oxygen species, elastase, lipopolysaccharide) cause extensive damage to the capillary plexus and alveolar endothelium. Such damage leads to the transudation of fluid into air spaces and subsequent exudate of neutrophils and protein-rich fluid, culminating in the formation of hyaline membranes that impair gas exchange and ultimately cause hypoxemic respiratory failure with local alveolar hypoxia. This respiratory failure contributes to the local hypoxia characteristic of COVID-19. In turn, hypoxia is a key promoter of angiogenesis. During SARS-CoV-2 infection, activation of the HIF-1α factor (hypoxia-inducible factor 1-alpha) is observed as a result of hypoxia associated with the observed inflammation. Activated HIF-1α regulates the expression of a wide range of genes involved in vasodilation, extracellular matrix remodelling, and angiogenic pathways (especially intussusceptive angiogenesis (IA)), such as VEGF (vascular endothelial growth factor), which additionally increases the permeability of blood vessels and, in a vicious circle mechanism, intensifies the cytokine storm [16,17,18,19,20].

Unfortunately, despite intensive research on the molecular mechanism of the infection caused by SARS-CoV-2, no effective treatment for COVID-19 has yet been developed. Nevertheless, to date, the World Health Organization has so far endorsed the safety and efficacy of AstraZeneca/Oxford (AZD1222), Johnson and Johnson/Janssen (Ad26.COV2.S), Moderna (mRNA-1273), Pfizer/BioNTech (BNT162b2), Sinopharm and Sinovac vaccines. AZD1222, Ad26.COV2.S and Gam-COVID-Vac (Sputnik V) contain DNA delivered in non-replicating recombinant adenoviral vector systems, while Pfizer and Moderna vaccines use mRNA technology and lipid nanoparticle delivery systems. All these vaccines contain material encoding the production of the (S) SARS-CoV-2 spike protein. However, previous clinical trials suggest that the COVID-19 vaccine from Pfizer may contribute to cardiovascular disorders. In short, SARS-CoV-2 vaccines can cause thromboembolic events such as cerebral vein thrombosis, and mRNA-based vaccines, in particular, can cause myocarditis/pericarditis [21]. Goh et al. (2022) showed that the Pfizer Comirnaty vaccine may contribute to the development of vaccine-induced immune thrombotic thrombocytopenia (VITT) [22]. Moreover, Cines’ teams (2021) found that central nervous system thrombosis may occur among patients who receive the Pfizer/BioNTech mRNA vaccine [23]. Interestingly, after the Pfizer/BioNTech vaccination, cases of myocarditis and pericarditis have been reported [24,25]. In addition, an analysis of the records of more than 2.5 million people given the Pfizer vaccine showed that the incidence of myocarditis was 2.3 per 100,000 people, rising to more than 10 per 100,000 in recipients aged 16 to 29 years [26]. On the other hand, a comparative study of people infected with the virus and post-vaccination showed that the risk ratio of myocarditis development is 5.39 in those who had been infected with SARS-CoV-2, whereas this ratio for vaccinated people is 1.27. Moreover, the risk factor of myocardial infarction was 1.07 for recipients of the Pfizer vaccine compared to 4.47 for those infected with SARS-CoV-2 [27].

Patone’s team analysed data on over 38 million people over 16 years of age from the English National Immunisation Management System (NIMS) regarding vaccinations against COVID-19 who received at least one dose of mRNA-1273 (n = 1,006,191), BNT162b2 (n = 16,993,389), or AZD1222 vaccine (n = 20,615,911). These studies showed that the risk of myocarditis was highest in the first seven days after receiving the first dose, and the increased risk could be described in all three vaccine groups. The mRNA-1273 vaccine showed the greatest increase in risk, with an incidence rate ratio (IRR) of 8.38, while those receiving the first dose of AZD1222 or BNT162b2 showed an IRR of 1.76 or 1.45, respectively. The risk of myocarditis was even higher after the second dose of both mRNA vaccines, with mRNA-1273 showing an IRR of 23.1 and BNT162b2 of 1.75, respectively [28]. Interestingly, additional analyses showed that the risk of myocarditis increased 16-fold after the second dose of the BNT162b2 vaccine and up to 44-fold after the second dose of mRNA-1273 vaccine in young men (16–19 years) compared to unvaccinated people during the same age [29,30]. Interestingly, in the case of the Ad26.COV2.S vaccine, only two cases of myocarditis after the first dose of Ad26.COV2 have been described so far. However, data on this vaccine are highly limited as no additional large-scale studies have been conducted to date beyond the pivotal Phase III study, and therefore, currently available data do not allow a clear conclusion on the incidence of myocarditis following Ad26.COV2 vaccination [31].

Similarly, in the case of the NVX-CoV2373 subunit vaccine (Novax) based on clinical trials, the Food and Drug Administration (FDA) has issued a warning about the increased risk of myocarditis and pericarditis in a fact sheet for healthcare providers [32]. On the other hand, the European Medicines Agency (EMA) did not issue a warning about the risk of myocarditis in the Summary of Product Characteristics (SmPC) [33]. During clinical trials of the NVX-CoV2373 vaccine, two cases of myocarditis were identified in 30,058 vaccinated people and one case among 19,982 people who received a placebo, which is associated with a statistically insignificant increase in the risk of myocarditis (relative risk 1.33, 95% CI 0.12–14.66). Additionally, three additional cases of myocarditis were identified in vaccinated patients during the clinical extension studies. Therefore, the FDA treated the results obtained with high caution, while the EMA considered the risk of myocarditis after NVX-CoV2373 vaccination with the preparation to be negligible [34,35].

Interestingly, for comparison, in 2019 (pre-pandemic period), it was estimated that 10.2–105.6 people had myocarditis per 100,000 inhabitants [36]. In turn, in the USA, the incidence of myocarditis unrelated to the SARS-CoV-2 infection after the outbreak of the pandemic was 150 cases/100,000 individuals. In contrast, the rate of myocarditis associated with the COVID-19 disease was 1000–4000 cases/100,000 individuals. Previous epidemiological studies have confirmed that SARS-CoV-2 infection increases the incidence of myocarditis by at least 15 times compared to the pre-pandemic level [37]. Therefore, the above data suggest that the incidence of cardiovascular disease after vaccination against COVID-19 is slightly increased compared to the unvaccinated population. Importantly, the incidence of cardiovascular disease varies between vaccines and depends on the number of doses received and the type of vaccination. However, despite intensive research on COVID-19 vaccines, the mechanism of vaccine-induced myocarditis remains unclear. Previous research indicates the existence of several mechanisms influenced by the immunological and genetic background, age, and gender. First, it may be due to mRNA immune reactivity; second, the spike protein antigen can induce cross-reaction of SARS-CoV-2 spike glycoproteins with myocardial contractile proteins; and finally, hyperimmunity related to hormones that contribute to the sex-specific differences seen in both mRNA vaccines against COVID-19 (higher risk in young men) [34,35].

Although complications from vaccines are exceedingly rare, they are extremely important and, in extreme cases, can be life-threatening. Interestingly, Wan et al. (2022) recently found a reduced risk of developing cardiovascular complications after COVID-19 infection in people vaccinated with BNT162b2 and/or CoronaVac [38].

Considering all the above reports, we aimed to investigate the impact of the SARS-CoV-2 infection and Pfizer Comirnaty vaccine on the molecular aspect of cardiovascular disorders, especially angiogenesis. Thus, in our study, we decided to evaluate the expression of those MMPs for which there is a well-documented association with vascular disease, endothelial dysfunction and vascular remodelling, as well as with thrombosis, including *HIF-1α*, *VEGFA*, *MMP-2* (matrix metalloproteinase 2), *MMP-7* (matrix metalloproteinase 7), *MMP-9* (matrix metalloproteinase 9), *TIMP1* (TIMP metallopeptidase inhibitor 1), and *ADAMTS1* (ADAM metallopeptidase with thrombospondin type 1 motif 1) [27,38,39,40]. The selection of the panel of genes whose expression was determined at the mRNA level was based on a review of the available literature and databases. The criteria for selecting the studied genes included their multifunctionality connection with the cytokine storm characteristic of the course of SARS-CoV-2 virus infection. As mentioned above, HIF-*1α* is a factor activated as a result of local hypoxia associated with damage to endothelial cells due to the intensification of inflammatory processes (cytokine storm) observed during SARS-CoV-2 infection. Then, HIF-*1α* can induce VEGFA, which is primarily responsible for increasing the permeability of the vascular endothelium to inflammatory mediators, enabling the spread of inflammation. On the other hand, the intensification of inflammation, including increased production of IL-lα, IL-1β, IL-6, and TNF-α (tumour necrosis factor α), activates the expression of metalloproteinases, especially MMP-2 and MMP-9, as well as ADAMTS proteins. The cytokine storm discussed above may be additionally amplified by the activity of prostaglandins, especially PGE2 (prostaglandin E2), activated by the NF-κB (nuclear factor kappa-light-chain-enhancer of activated B cells) pathway. In turn, active PGE2 can block the IFN-β (interferon β), which downregulates MMP-9 production and upregulates tissue inhibitors of matrix metalloproteinases I (TIMP) (MMP-9 inhibitor). Moreover, many previous studies confirmed that levels of VEGFA, MMP-2, MMP-9, and ADMTS1 were altered in the local tissue and plasma of patients with COVID-19 [15,16,17,18,19,20]. However, despite numerous studies, the literature does not indicate the impact of SARS-CoV-2 infection on the level of mRNA expression of the analysed genes in peripheral blood mononuclear cells (PBMCs). Additional information about all studied gene products is presented in Table 1.

## 2. Results

### 2.1. mRNA Expression of HIF-1α, VEGFA, MMP-2, MMP-7, MMP-9, TIMP1, and ADAMTS1

The real-time PCR analysis (Figure 1A–G) showed that convalescents vaccinated with one dose of BNT162b2 were characterised by higher *MMP-7* expression than non-vaccinated convalescents and healthy volunteers vaccinated with one dose of BNT162b2 (Figure 1D; *p* < 0.05). Moreover, non-vaccinated convalescents showed increased mRNA expression of *ADAMTS1* compared to healthy volunteers vaccinated with one dose of BNT162b2 (Figure 1G; *p* < 0.05). In the case of the other analysed genes, no significant differences were found (Figure 1A–C,E,F).

### 2.2. Influence of Age, Gender, and Co-Existing Diseases on the mRNA Expression of All Studied Genes

Two-way ANOVA analysis (Figure 2A–G) showed that the non-vaccinated convalescents exhibited increased mRNA expression of *ADAMTS1* compared to healthy volunteers vaccinated with one dose of BNT162b2 (*p* = 0.018). The expression of *ADAMTS1* (Figure 2G) did not show statistically significant differences in the indicators depending on gender (*p* > 0.05). Interestingly, statistical analysis indicated that vaccinated convalescents were characterised by higher *MMP-7* expression than non-vaccinated convalescents and vaccinated healthy volunteers. Moreover, two-way ANOVA analyses revealed significant gender and COVID-19/BNT162b2 vaccination effects on *MMP*-7 mRNA levels. Overall, *MMP-7* mRNA expression (Figure 2D) was higher in the vaccinated convalescent group than in non-vaccinated convalescents (*p* = 0.014) and vaccinated healthy volunteers (*p* = 0.025) in the female subgroups. The expression of *MMP-7* did not demonstrate statistically significant differences between studied groups in male subpopulations (*p* > 0.05). In the case of *HIF-1α*, *VEGFA*, *MMP-2*, *MMP-9*, and *TIMP1,* no significant differences were found (Figure 2A–C,E,F).

In the case of *ADAMTS1* and *MMP-7*, two-way ANOVA analyses showed no statistically significant differences between all studied groups depending on co-existing diseases and the age of the participants (Appendix A).

## 3. Discussion

So far, most attention has been attributed to inflammation in the pathophysiology of COVID-19 due to its rampant form known as a cytokine storm [41,42]. Nevertheless, some studies highlight the role of other biochemical pathways, such as hypercoagulability and pathological angiogenesis driven by endothelial cell dysfunction in patients with SARS-CoV-2 infection [43,44]. However, even though there is widespread interest in the pathophysiology of the COVID-19 disease, relatively little is known about the associated molecular changes in the epithelial function of patients who suffer from COVID-19 and patients after vaccination against SARS-CoV-2 [45]. The aim of the presented study was thus to assess the impact of SARS-CoV-2 virus infection on the mRNA expression level of genes involved in the angiogenesis process, including *HIF-1α, VEGFA, MMP-2, MMP-7, MMP-9, TIMP1*, and *ADAMTS1*. Our study is the first study to analyse mRNA expression levels of genes associated with angiogenesis in COVID-19 convalescents and BNT162b2 vaccinated individuals.

Abnormal angiogenesis in the course of COVID-19 is associated with endothelial dysfunction [46]. COVID-19-related endothelial dysfunction is characterised by acute vasculitis and recruitment of perivascular T cells, leading to disruption of the alveolar–capillary barrier and increased permeability. Endothelial cells surrounded by T lymphocytes show signs of strong activation, termed “endothelial inflammation” [47]. The most crucial factor of “endothelial inflammation” in the pathogenesis of COVID-19 is the cytokine storm that initiates a series of changes that may lead to the development of multi-organ failure and even death [48]. Infection of endothelial cells by SARS-CoV-2 causes swelling and damage to endothelial cell barriers and abnormal microvascular architecture. Vascular damage is accompanied by thrombosis, vasoconstriction and marked IA, which is a unique rapid process of blood vessel neoformation involving the division of the vessel into two lumens as a result of the inclusion of circulating angiogenic cells [47]. The primary inducer of the mentioned cytokine storm is the increased production of HIF-1α in response to hypoxia accompanying SARS-CoV-2 infection [49,50]. Moreover, HIF-1α may enhance the expression of *VEGF*, which plays a key role in angiogenesis in response to hypoxia, activating vascular growth and improving oxygen supply, which in turn reduces tissue hypoxia, and prevents cell death, macrophage migration, and inflammation. On the other hand, VEGF increases vascular permeability and, thus, the migration of immune cells into the tissues, which exacerbates inflammation and cytokine storm [51]. Moreover, previous studies showed that capillaries in the lungs of COVID-19 patients show cylindrical microstructures in the capillary lumen, indicating activation of IA in the course of SARS-CoV-2 infection. Interestingly, the IA induction may be a consequence of the overexpression of VEGF and angiopoietins-1/2, which may lead to an increase in the number of small holes in primordial capillary plexuses in patients with COVID-19 [16,17,47]. Despite numerous studies pointing to the significant role of HIF-1α and VEGFA, our studies did not show significant differences in mRNA expression for these factors in all studied groups.

Previous clinical studies have shown that the cytokine storm observed in the course of SARS-CoV-2 infection is a result of the overproduction of different pro-inflammatory cytokines produced by macrophages and epithelial cells as a response to the innate immune system activation, such as interleukin 1β (IL-1β), interleukin 6 (IL-6), and TNF-α [41,52]. This overproduction of pro-inflammatory cytokines, apart from hypoxia and oxidative stress, activates the angiogenesis cascade, leading to the dysfunction of endothelial cells [53]. Consequently, an autopsy confirmed the presence of characteristic vascular lesions in the lungs of COVID-19 patients, consisting of severe endothelial damage due to the presence of intracellular viruses. In addition, histological analysis of pulmonary vessels in COVID-19 patients revealed extensive thrombosis with microangiopathy. Interestingly, microblots in the alveolar capillaries were nine times more common in patients with COVID-19 than in patients with influenza [54]. Therefore, among our analysed genes were those encoding proteins belonging to the family of metalloproteinase enzymes. A common feature of MMPs is the presence of zinc in their catalytic site for activity and their synthesis as inactive zymogens. Typically, MMPs are only expressed when and where they are needed for tissue remodelling [55]. Transcription can be induced by a variety of signals, including cytokines, growth factors, and mechanical stress [56]. Therefore, the SARS-CoV-2 virus infection state may contribute to increased expression of MMPs. Previous studies confirmed that patients with COVID-19 were characterised by increased protein expression of MMP-1, MMP-2, MMP-7, MMP-8, and MMP-14 in lung tissue as compared to controls who tested negative for COVID-19 [57]. In turn, we found that mRNA expression of *MMP-7* was higher in COVID-19 convalescents vaccinated with one dose of BNT162b2 than in COVID-19 convalescents and healthy volunteers vaccinated with one dose of BNT162b2. These data suggest that the history of COVID-19 combined with vaccination with Pfizer’s preparation is a factor significantly influencing the intensification of processes related to microarray remodelling and angiogenesis by the VEGF pathway, which are mainly under the control of MMP-7. Moreover, we found that COVID-19 convalescents were characterised by higher *ADAMTS1* expression than healthy volunteers vaccinated with one dose of BNT162b2. Unlike MMP-7, ADAMTS1 has anti-angiogenic properties. Therefore, the obtained data may suggest that in recovered patients, mechanisms are activated to restore the pre-infection state, which means that the mechanisms related to remodelling and angiogenesis activated during the cytokine storm are blocked by angiogenesis inhibitors, including TIMPS. Interestingly, a growing body of evidence suggests that MMPs and metalloproteinase disintegrins (ADAMs) are responsible for the excretion of growth factors (e.g., heparin-binding epidermal growth factor), thereby transactivating related growth factor receptors (e.g., EGFR, epidermal growth factor receptor) in the development of hypertrophy associated with arterial hypertension [58]. Therefore, SARS-CoV-2 infection may contribute to the development of arterial hypertension. Clinical trials support this hypothesis. Akpek (2022) showed that both systolic and diastolic blood pressure were significantly higher in the post-COVID-19 period than in the hospital admission period [59]. In addition, in the case of *MMP-7* expression, we found differences in expression in the group of COVID-19 convalescents and healthy volunteers after vaccination, depending on gender. In both studied groups, men showed higher levels of *MMP-7* expression than women. Similarly, other studies to date have confirmed the increased expression of MMPs in men with various pathological conditions. The Swedes showed higher levels of MMP-3 in men than in women with myocardial infarction and in people in the control group [60]. An independent study conducted in the United States showed similar results in patients with various rheumatic diseases and in healthy controls [61]. Other authors also confirmed higher levels of MMP-3, MMP-8, and MMP-9 in British men with rheumatoid arthritis compared to women [62]. Another study found higher levels of MMP-8 in Peruvian men with tuberculosis [63]. The causes of this imbalance between men and women are unclear but may be the result of a differentiated hormone balance. The results so far suggest that males exhibit a more aggressive and detrimental inflammatory immune response to microbial stimuli, including viruses, than females of childbearing age [64]. Interestingly, it was also shown that decreased production of MMP-9 and TNF-α by neutrophils in the course of infection was also observed in women in the periovulatory period when oestrogen levels were higher [65]. Moreover, women generate more robust and potentially protective humoral and cell-mediated immune responses following antigenic challenge than their male counterparts. Previous studies confirmed that women were characterised by higher circulating levels of IgM than men [64].

On the other hand, the clinical trials cited in the Section 1 of this manuscript [17,18,19,20,21,22,23,24,25,26,27,28,29,30,31,32,33,34,35,36,37] also indicate a significant role of the Pfizer Comirnaty vaccine in the development of cardiovascular disease, including those related to the abnormal course of angiogenesis. However, our study is the first to demonstrate the potential impact of Pfizer’s vaccine on the angiogenesis pathway at the molecular level. Our results indicate that convalescents vaccinated with one dose of the preparation had a higher level of *MMP-7* expression than COVID-19 convalescents and healthy volunteers after vaccination. Our research confirms the previous results indicating the role of MMPs in post-inflammatory remodelling of the myocardium in myocarditis [66,67].

We have first demonstrated that SARS-CoV-2 infection and BNT162b2 vaccination impact the mRNA expression levels of genes associated with angiogenesis. However, we are aware that the presented study has some limitations. Our study is preliminary, limited to a relatively small sample size and homogeneous ethnicity of the participants, which in turn gives the possibility that the results may not be duplicated in other populations. Therefore, our results must be treated with great caution and cannot be freely extrapolated to other ethnic groups. However, in the future, our study requires studies on larger numbers of patients. Moreover, based on the results obtained, we cannot clearly determine that the change in *MMP-7* and *ADAMTS1* expression is independent of other factors. However, the literature data suggest that the disorders we observe may be the result of the cytokine storm observed in the course of COVID-19. Moreover, we cannot clearly determine whether the changed expression at the mRNA level translates into changes in the expression of the appropriate proteins encoded by the genes. Additionally, altered *MMP-7* and *ADAMTS1* expression in PBMCs should be analysed with caution because changes observed in PBMCs may not reflect local changes in the lung and other affected tissues. Therefore, it is necessary to conduct further research, also considering the panel of genes encoding pro-inflammatory cytokines responsible for the cytokine storm and the determination of the level of angiogenesis-related proteins.

## 4. Materials and Methods

### 4.1. Participants

In this research, the participants were as follows: 33 COVID-19 convalescents having a positive test result for SARS-CoV-2 infection, as indicated by quantitative reverse-transcription polymerase chain reaction of nasopharyngeal swab samples; 35 healthy people vaccinated with one dose of BNT162b2; 19 convalescents (with a confirmed status by test) vaccinated with one dose of BNT162b2. All patients were from inpatient and outpatient units of Rehabilitation Division III General Hospital in Lodz, Poland. SARS-CoV-2 infection was confirmed as a positive test result based on the reverse-transcription polymerase chain reaction (rt-PCR) of nasopharyngeal swab samples. In the case of healthy volunteers vaccinated with one dose of BNT162b2, all subjects had negative tests for the SARS-CoV-2 infection. Furthermore, each volunteer enrolled in the study was a native Pole from central Poland (not related to one another), randomly selected without replacement sampling, and all studied groups were matched by age and gender. Moreover, subjects with cancer were excluded from the study. Participation in this presented study was voluntary. All patients were informed about the details of the study and confidentiality, as well as assured of their voluntary participation in the experiment. Finally, participants gave their written informed consent to contribute to this experiment. After signing the consent to participate in the study, whole blood from the antecubital vein was collected from 1 to 4 weeks after vaccination and, in the case of other people, 3 days after admission to the hospital. All procedures were carried out according to the Helsinki Declaration and were approved by the Ethics Committee of the Medical University of Lodz, Poland No. RNN/101/22/KE and the Bioethics Committee of the Faculty of Biology and Environmental Protection of the University of Lodz, Poland 16(I)/KBBN-UŁ/I/2021-22. Characteristics of participants are presented in Table 2.

### 4.2. Blood Sample Collection and RNA Isolation

In the present study, samples of peripheral venous blood were collected from each participant into 5 mL vacutainers with EDTA, and then all blood samples were coded and stored at −20 °C until further use. Then, total RNA samples were extracted from frozen whole blood samples using DNA/RNA Extracol Kit (EURx, Gdansk, Poland) according to the manufacturer’s instructions. The purity and relative RNA concentration were determined with Bio-Tek Synergy HT Microplate Reader (Bio-Tek Instruments, Winooski, VT, USA). The RNA sample purity was selected with a yield of a 260/280 ratio near 1.8–2.0 [68]. Finally, all RNA samples were stored at −20 °C until use.

### 4.3. cDNA Synthesis mRNA

Total RNA samples (10 ng/µL) were reverse transcribed with a High-Capacity cDNA Reverse Transcription Kit (Applied Biosystems, Foster City, CA, USA). All components were mixed to form a 20 µL reaction volume: nuclease-free water; 10xRT Buffer; 10xRT Random Primers; 25xdNTP Mix (100 mM); total RNA (0.5 ng/µL) and MultiScribe^®^ Reverse Transcriptase. The program is as follows: 10 min at 25 °C, 120 min at 37 °C, and 5 min at 85 °C. Reverse transcription polymerase chain reaction (rt-PCR) was performed in a C1000™ programmed Thermal Cycler (Bio-Rad Laboratories Inc., Hercules, CA, USA). After the reverse transcription, the cDNA samples were stored at −20 °C until further analysis.

### 4.4. mRNA Expression Levels

The real-time PCR was used to the evaluation of mRNA expression and performed using RT PCR Mix Probe (A&A Biotechnology, Gdansk, Polska) and species-specific TaqMan Gene Expression Assay (Thermo Fisher Scientific, Waltham, MA, USA). Expression of *HIF-1α* (assay ID: Hs00936368_m1), VEGFA (assay ID: Hs00900055_m1), *MMP-2* (assay ID: Hs01548727_m1), *MMP-7* (assay ID: Hs01042796_m1), *MMP-9* (assay ID: Hs00957562_m1), *TIMP1* (assay ID: Hs01092511_m1) and *ADAMTS1* (assay ID: Hs00199608_m1) genes was performed on a TaqMan Gene Expression Assay in a CFX96™ Real-Time PCR Detection System Thermal Cycler (Bio-Rad Laboratories Inc.). The housekeeping gene for the human *18S* ribosomal RNA gene (18S; Hs99999901_s1) was used as an internal control (reference gene), as it normalizes RNA input measurement errors and variations in rt-PCR efficiency. All the samples were performed in duplicates. The PCR amplification program was the following: 95 °C 3 min (enzyme activation), 50 cycles of 95 °C 30 s (denaturation) and 60 °C 60 s (annealing/extension). The cycle threshold (Ct) values were calculated automatically by a CFX96 Real-Time PCR Detection System Software System (Bio-Rad Laboratories, Inc., Hercules, CA, USA). For each sample, the gene expression of the target mRNA was calculated relative to a reference gene (ΔCt sample = Ct _target gene_ − Ct _reference gene_). Levels of gene expression are given as a normalisation ratio calculated as fold = 2^−ΔCt^ sample [69].

### 4.5. Statistical Analysis

Statistics were calculated using Statistica 12 (StatSoft, Hamburg, Germany), SigmaPlot 11.0 (Systat Software Inc., San Jose, CA, USA) and GraphPad Prism 5.0 (GraphPad Software, Inc., San Diego, CA, USA). Data are expressed as the mean ± standard deviation. Then, the one-way analysis of variance (ANOVA) was used to detect significant differences between samples with normal distribution, whereas differences between probes with non-normal distribution were confirmed by the Kruskal–Wallis test. Finally, Dunn’s test was used as a post-hoc test. Moreover, data regarding the effects of gender/age/comorbidities and SARS-CoV-2 infection/BNT162b2 vaccination on mRNA expression of all studied genes were analysed using two-way ANOVA analyses. Finally, the Bonferroni test was used as a post-hoc test. *p* values < 0.05 were considered significant.

## 5. Conclusions

Our study confirms the hypothesis that SARS-CoV-2 infection and vaccination with the Pfizer preparation affect the regulation of angiogenesis at the molecular level by modulating the expression of MMP-7 and ADAMTS1 genes. Moreover, the observed changes are sex-dependent. This knowledge can significantly contribute to the development of effective targeted therapeutic strategies in the future. However, keep in mind that more research is needed.

## Figures and Tables

**Figure 1 ijms-24-16094-f001:**
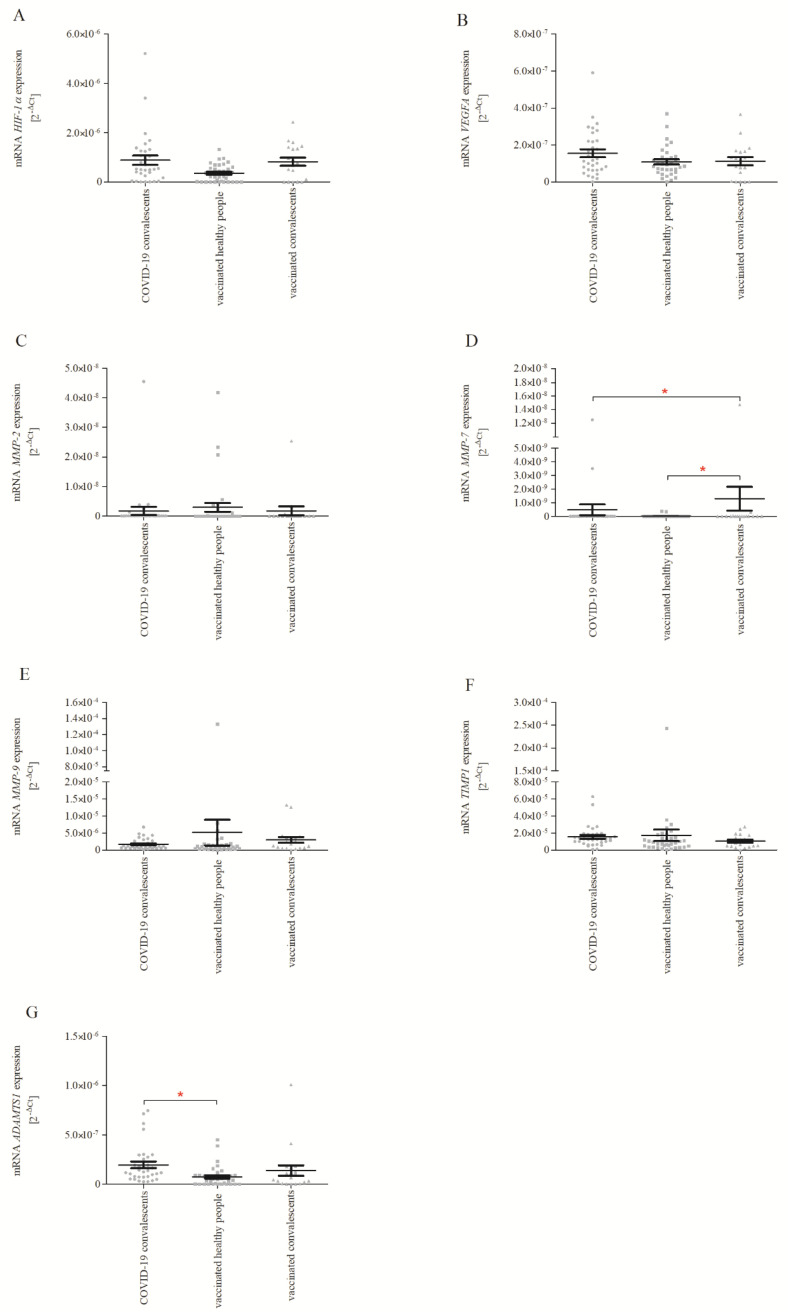
mRNA expression of *HIF−1α* (**A**), *VEGFA* (**B**), *MMP−2* (**C**), *MMP−7* (**D**), *MMP−9* (**E**), *TIMP1* (**F**) and *ADAMTS1* (**G**) in the COVID-19 convalescents, healthy people vaccinated with BNT162b2, convalescents vaccinated with one dose of BNT162b2. Relative gene expression levels were estimated using a 2^−ΔCt^ (Ct _target gene_ − Ct _18S_) method. Data represent means ± SD. * *p* < 0.05.

**Figure 2 ijms-24-16094-f002:**
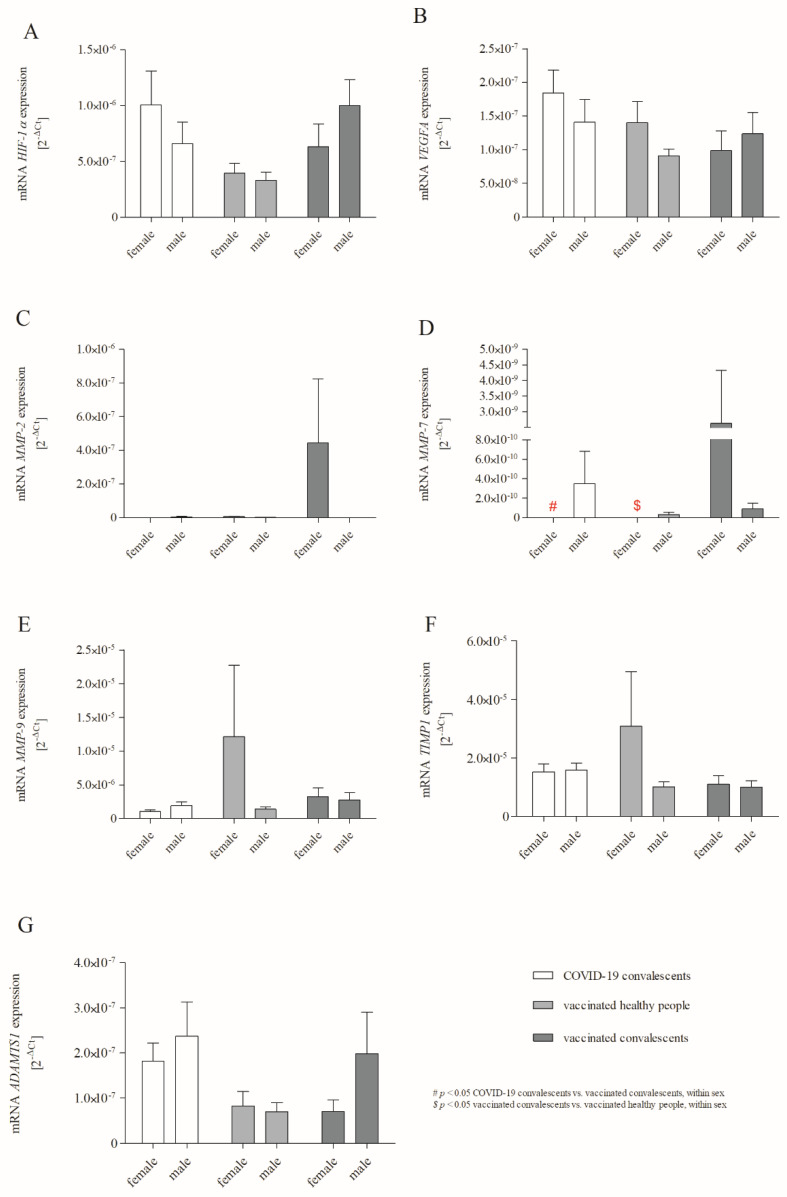
Two-way ANOVA shows significant effects of gender and COVID-19/BNT162b2 vaccinations on studied gene expression, including *HIF-1α* (**A**), *VEGFA* (**B**), *MMP-2* (**C**), *MMP-7* (**D**), *MMP-9* (**E**), *TIMP1* (**F**) and *ADAMTS1* (**G**). Relative gene expression levels were estimated using a 2^−ΔCt^ (Ct _target gene_ − Ct _18S_) method. Data represent means ± SD. # *p* < 0.05 COVID-19 convalescents vs. vaccinated convalescents, within sex; $ *p* < 0.05 vaccinated convalescents vs. vaccinated healthy people, within sex.

**Table 1 ijms-24-16094-t001:** Characteristic of all studied genes *.

Angiogenesis				
Gene	Protein	Gene Location	Function	Tissue mRNA Expression
*HIF-1α*	Hypoxia-inducible factor 1 subunit alpha	14q23.2	The primary transcriptional regulator of the adaptive response to hypoxia. Under hypoxic conditions, it activates the transcription of over forty genes, including erythropoietin, glucose transporters, glycolytic enzymes, vascular endothelial growth factor, HILPDA, and other genes whose protein products increase oxygen delivery or facilitate metabolic adaptation to hypoxia. HIF-1 thus plays an essential role in embryonic vascularization, tumour angiogenesis and pathophysiology of ischemic disease.	Detected in all tissue
*VEGFA*	Vascular endothelial growth factor A	6p21.1	Growth factor active in angiogenesis, vasculogenesis and endothelial cell growth. Induces endothelial cell proliferation, promotes cell migration, inhibits apoptosis, and induces permeabilization of blood vessels.	Detected in all tissue
*MMP-2*	Matrix metalloproteinase 2	16q12.2	It is a ubiquitous metalloproteinase that is involved in the remodelling of the vasculature, angiogenesis, tissue repair, tumour invasion, inflammation, and atherosclerotic plaque rupture, as well as degrading extracellular matrix proteins. Moreover, MMP-2 can also act on several nonmatrix proteins such as big endothelial one and beta-type CGRP promoting vasoconstriction.	Detected in all tissue
*MMP-7*	Matrix metalloproteinase 7	11q22.2	This protein degrades casein, gelatins of types I, III, IV, and V, and fibronectin, whereas it activates procollagenase. Moreover, MMP-7 modulates the VEGF pathway in endothelial cells, degrading soluble VEGFR-1 and, in turn, promoting angiogenesis. MMP-7 also enhances endothelial cell proliferation.	Detected in all tissue
*MMP-9*	Matrix metalloproteinase 9	20q13.12	MMP-9 plays an essential role in local proteolysis of the extracellular matrix and in leukocyte migration. MMP-9 also promotes endothelial cell migration and triggers the angiogenic switch by releasing VEGF. Moreover, MMP-9 may be associated with the development of vein thrombosis.	Group enriched (bone marrow, lymphoid tissue)
*TIMP1*	TIMP metallopeptidase inhibitor 1	Xp11.3	TIMP1 forms the complex with targets metalloproteinases, such as collagenases, and irreversibly inactivates them by binding to their catalytic zinc cofactor. Acts on MMP-1, MMP-2, MMP-3, MMP-7, MMP-8, MMP-9, MMP-10, MMP-11, MMP-12, MMP-13, and MMP-16. TIMP1 regulates cell differentiation, migration, and cell death. TIMP1 is a well-documented inhibitor of apoptosis and blocks the endothelial cell response to angiogenic factors, e.g., basic fibroblast growth factor (bFGF). Moreover, TIMP1 may be associated with prothrombotic state.	Detected in all
*ADAMTS1*	ADAM metallopeptidase with thrombospondin type 1 motif 1	21q21.3	ADAMTS1 cleaves aggrecan and a cartilage proteoglycan. It also has angiogenic inhibitor activity.	Detected in all

* The data in the table were developed based on the following databases: UniProtKB, Gene and The Human Protein Atlas.

**Table 2 ijms-24-16094-t002:** Characteristics of study participants.

Characters		COVID-19 Convalescents (n = 33)	Healthy People Vaccinated with One Dose of BNT162b2 (n = 35)	Convalescents Vaccinated with One Dose of BNT162b2 (n = 19)
Number of women		15	12	9
Number of men		18	23	10
Age (mean ± SD)		63.29 ± 8.62	67.37 ± 12.05	68.58 ± 6.70
Co-existing diseases	Non	22	9	5
	Atherosclerosis	2	2	1
	Diabetes	5	6	4
	Hypertension	9	24	13
	Stroke	12	12	13
	Paresis	10	11	10
	Hypercholesterolemia	6	0	0
	Heart failure	0	1	0

## Data Availability

The data supporting this study’s findings are available on request from the corresponding author [Elżbieta Miller: elzbieta.dorota.miller@umed.lodz.pl].

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
