# Peer review of "Effect of SARS-CoV-2 Infection and BNT162b2 Vaccination on the mRNA Expression of Genes Associated with Angiogenesis"

_ijms, 2023, doi:10.3390/ijms242216094_

Round 1

Reviewer 1 Report

Comments and Suggestions for Authors

In this work the authors carried out a genetic expression analysis of genes associated with the angiogenesis profile of convalescent vaccinated individuals with the mRNA vaccine from Pfizer in addition to SARS-CoV-2 infection. In this work they demonstrate that SARS-CoV-2 infection and vaccination promote a high level of MMP7. Likewise, it is found that convalescent individuals have high levels of expression of ADAMTS1.

In addition, it is reported that SARS-CoV-2  may be related to immunological problems such as cytokine storm, inflammation, extracellular traps by neutrophils, which if persisted can exacerbate tissue damage, including organs where viral replication is not present and can cause hypercoagulation and angiogenesis.

Side effects of messenger RNA vaccines are also mentioned including thrombotic events and cerebral vein thrombosis, myocarditis, and pericarditis, as well as immune thrombotic thrombocytopenia.

The article remarks the risk of myocarditis and thrombosis due to vaccination and/or SARS-Co_2 infection; however, it does not mention the risk of myocarditis and thrombosis in populations that were not vaccinated nor infected (for instance, before this pandemics). So, what is the risk of myocarditis for non-vaccinated/non-infected people? This really would be an excellent comparison.

The risk of myocarditis in people vaccinated with vaccines other than mRNA was not reported in this work, so, it is not clear whether myocarditis is due to vaccines containing Spike or mRNA vaccines.

Nowadays, there is a lot of information about hypercoagulability, but the pathogenesis of this condition in convalescent patients is not explained. Information on the pathogenesis of coagulopathies in COVID-19 should be reviewed.

In this work, it will be important to emphasize that the increase of mRNAs related to angiogenesis is not necessarily related to the increase of the related proteins.

Finally, it would be more appropriate to mention whether angiogenesis is positively or negatively regulated by MMP7 and ADAMTS1.

Comments on the Quality of English Language

Abbreviations and acronyms must be reviewed.

Reviewer 2 Report

Comments and Suggestions for Authors

The work "Effect of SARS-CoV-2 infection and BNT162b2 vaccination on the mRNA expression of genes associated with angiogenesis " was written at the end of 2022 .The presented research aimed to verify the impact of the SARS-CoV-2 infection and Pfizer Comirnaty vaccine on the molecular aspect of angiogenesis. At the time of writing this manuscript (until 9 November 2022), more than 638 million cases of SARS-CoV-2 (severe acute respiratory syndrome coronavirus 2) infection have been recorded in 192 countries and territories. Of this number, there are almost 13 million active cases, more than 618 million warmings and more than 6.68 million deaths.

This study is the first to analyse the mRNA expression levels of genes associated with angiogenesis in individuals convalescing from COVID-19 and vaccinated with BNT162b2. However more controls are needed.

The study needs a review of basic terms such as (jev) written in the text when  talks about the vaccines I believe to be  Janssen (Johnson & Johnson) COVID-19 Vaccine.

Reviewer 3 Report

Comments and Suggestions for Authors

Given that thromboembolic events and myocarditis/pericarditis have been reported to occur occasionally following the administration of SARS-CoV-2 vaccines such as BNT162b2, the authors sought to investigate the effects of SARS-CoV-2 infection and BNT162b2 vaccination on the mRNA expression of genes associated with angiogenesis.  Based on real-time PCR analysis of gene expression in peripheral blood cells of patients, they report higher matrix metalloproteinase MMP-7 expression in convalescents vaccinated with one dose of BNT162b2 and increased expression of the angiogenic inhibitor ADAMTS1 in non-vaccinated convalescents compared to control groups.

Although the effects of SARS-CoV-2 infection and/or BNT162b2 vaccination on thromboembolic events and myocarditis/pericarditis are clinically significant, the experimental design of the study raises important questions.  As shown in Table 2, a large proportion of the study participants have underlying diseases known to alter endothelial cell and vascular functions (e.g., stroke, hypertension, diabetes, atherosclerosis, hypercholesterolemia), which renders this patient cohort less than ideal to study the impact of SARS-CoV-2 infection and/or BNT162b2 vaccination on angiogenesis.  Additionally, the rationale for selecting the studied genes among many other genes of interest is not clearly defined, and some of the studied genes have multiple functions other than angiogenesis, such as MMP-7 is involved in tissue injury/repair, fibrosis and tumor invasion.

Line 18 and 84: since you claim that this is the first study to analyze mRNA expression of genes associated with angiogenesis in Covid-19 convalescents and BNT162b2 vaccinated individuals, the words “verify” and “verification” must be changed to, for example, “investigate” and “investigation”.  The former words suggest that you are merely confirming previous results.

Line 34: update your manuscript with current information and references, and remove mention of a specific date.

Line 42, 138-139, 157-158: provide supporting references.

Table 1: which are the functions of MMP-7, MMP-9 and TIMP9 related to endothelial cells and angiogenesis?  Which of the studied genes are associated with hypercoagulable state and/or thromboembolism?

Line 124-126: are the “comorbidities” the same as the “co-existing diseases” in Table 2?  Please provide the results of this two-way ANOVA.

Line 183-184: what do you mean by “the SARS-CoV-2 virus infection state appears to be the ideal cause of increased expression of MMPs”.  How can you be sure of that? 

Line 237-241: describe the study participants in more detail.  How long has been the convalescence period (indicate range) before blood sample collection? Was a commercial test or a laboratory developed SARS-CoV-2 test(s) used?  Did the “healthy people” test negative?  How long after the BNT162b2 vaccination was the blood sample collected (indicate range)?

Figure 1 and 2, Table 2, line 295: be consistent with the use of standard deviation or standard error of the mean throughout the paper.

Round 2

Reviewer 1 Report

Comments and Suggestions for Authors

Abbreviations should be improved.

The increase in the percentage of cases of pre- and post-pandemic myocarditis (65.40%) should be reported as incidence.

Reviewer 3 Report

Comments and Suggestions for Authors

Overall, the revised text of this manuscript is not clearly written.  Ideas are not well defined and articulated, and do not flow naturally.  

Introduction: the text of your introduction is much too long and lacks a clear focus.  Clarify the rationale and the experimental design of the study.

Line 72-73: clarify the sentence “SARS-CoV-2 infection may lead to a hyper-coagulation state and pathologic angiogenesis attributing complications, including myocarditis.”  Present a clear hypothesis linking SARS-CoV-2 infection, endothelial infection and dysfunction, coagulopathy, myocarditis and angiogenesis repair.  See references below.

Line 454-473: your justification of the selection of the studied genes does not make sense, I’m sorry to say.  You begin by stating that your criteria have to do with their multifunctionality, involvement with the cytokine storm and gene mutations/polymorphisms.  You do not provide convincing evidence for the first two criteria. What were the inclusion and exclusion criteria for selecting the genes in Table 1?  Are these genes associated with the cytokine storm? You did not consider the physiological changes in gene expression associated with angiogenesis; instead you refer for no clear reason to gene mutations/polymorphisms and do not discuss whether these alterations are associated with vascular changes.  Also you don’t provide evidence of gene alterations in your patient cohort.  This section should be moved to the end of the introduction.

Discussion: by which mechanism(s) is the cytokine storm associated with angiogenesis?  Provide detailed supporting evidence.

Your review of the pertinent literature is incomplete. Here are a few examples:

Restrepo MI, et al. Cardiovascular Complications in Coronavirus Disease 2019-Pathogenesis and Management. Semin Respir Crit Care Med. 2023 Feb;44(1):21-34. doi: 10.1055/s-0042-1760096.

Mentzer SJ, et al. Endothelialitis, Microischemia, and Intussusceptive Angiogenesis in COVID-19. Cold Spring Harb Perspect Med. 2022 Oct 3;12(10):a041157. doi: 0.1101/cshperspect.a041157.

Madureira G, Soares R. The misunderstood link between SARS-CoV-2 and angiogenesis. A narrative review. Pulmonology. 2023 Jul-Aug;29(4):323-331. doi: 10.1016/j.pulmoe.2021.08.004.

Smadja DM, et al. COVID-19 is a systemic vascular hemopathy: insight for mechanistic and clinical aspects. Angiogenesis. 2021 Nov;24(4):755-788. doi: 10.1007/s10456-021-09805-6.
